# Systemic Sclerosis in Kazakh Patients: A Preliminary Case–Control Immunogenetic Profiling Study

**DOI:** 10.3390/pathophysiology32040057

**Published:** 2025-10-28

**Authors:** Lina Zaripova, Abai Baigenzhin, Alyona Boltanova, Zhanna Zhabakova, Maxim Solomadin, Larissa Kozina

**Affiliations:** 1JSC National Scientific Medical Center, 42 Abylai Khan Ave., Astana 010009, Kazakhstan; a.baigenzhin@nnmc.kz (A.B.); alyona.boltanova@gmail.com (A.B.); zhanna_zhabakova@mail.ru (Z.Z.); l.kozina@nnmc.kz (L.K.); 2NCJSC International Center for Vaccinology, Kazakh National Agrarian Research University, 8 Abay Ave., Almaty 050010, Kazakhstan; maks50@gmail.com

**Keywords:** systemic sclerosis, genes, antibodies, interleukin-6, Kazakh cohort

## Abstract

**Background/Objectives**: Systemic sclerosis (SSc) is a heterogeneous connective tissue disease characterized by immune dysregulation, vasculopathy, and fibrosis. Objectives: To evaluate the genetic architecture and autoantibody profile in a Kazakh cohort of patients with SSc. **Methods**: A total of 26 Kazakh patients with diffuse SSc were examined for disease activity and organ impairment using EScSG and the modified Rodnan skin score (mRSS). Eighteen healthy volunteers were enrolled in the control group. Antinuclear factor (ANF) was estimated on HEp-2 cells, while antibodies to Scl-70, CENP-B, U1-snRNP, SS-A/Ro52, SS-A/Ro60, Sm/RNP, Sm, SS-B, Rib-P0, and nucleosomes were determined by immunoblotting. The level of IL-6 cytokine was detected using ELISA. To investigate the genetic basis of SSc in Kazakh patients, a custom AmpliSeq panel including targeting immune/fibrosis pathways and 120 genes was used on the Ion Proton sequencer. The statistical analysis of categorical variables was conducted using Fisher’s exact test and Chi-square (χ^2^) test. **Results**: The examination of SSc patients (mRSS 16 ± 7.2; EScSG 3.54 ± 2.18) revealed a broad range of antibodies to Scl-70, CENP-B, SS-A/Ro60, SS-A/Ro52, U1-snRNP, and RNP/Sm, which were undetectable in the control group. Genetic analysis identified multiple variants across immune regulatory genes, including likely pathogenic changes in SAMD9L, REL, IL6ST, TNFAIP3, ITGA2, ABCC2, AIRE, IL6R, AFF3, and TREX1. Variants of uncertain clinical significance were detected in LY96, IRAK1, RBPJ, IL6ST, ITGA2, AIRE, IL6R, JAZF1, IKZF3, IL18, IL12B, PRKCQ, PXK, and DNASE1L3. Novel variants at the following genomic coordinates were identified and have not been previously reported in association with SSc: LY96 (chr8:74922341 CT/C), PTPN22 (chr1:114381166 CT/C), IRAK1 (indels at chrX:153278833), and SAMD9L (chr7:92761606 GT/G; chr7:92764981 T/TT). **Conclusions**: The first immunogenetic investigation of SSc in Kazakhstan revealed a polygenic architecture involving immune signalling pathways that partially overlap with international cohorts while exhibiting region-specific variation. Although the limited sample size and lack of functional validation constrain the interpretability of the findings, the results provide a framework for larger research to confirm the pathogenic mechanisms and establish clinical relevance.

## 1. Introduction

Systemic sclerosis (SSc) is a complex connective tissue disorder with an unknown etiology, characterized by dysfunction in the microcirculation, the disruption of the immune system, and the formation of fibrous tissue in the skin and internal organs [1].

The incidence of SSc varies considerably worldwide, ranging from 3.1 per 100,000 to 144.5 per 100,000 among specific ethnic groups. This condition predominantly affects middle-aged women, with a cumulative incidence of 33.9 per 100,000, compared with men, who have an incidence of 6.0 per 100,000 [2]. Research has shown that, while men are less likely to develop SSc, they do experience a more severe course of the disease and have a higher risk of premature mortality compared with women [3,4]. SSc has the highest mortality rate among rheumatic disorders [5]. Currently, no national registry or epidemiological database are available in Kazakhstan, and thus the incidence data presented here reflect global and regional estimates rather than local statistics.

The cause of SSc remains obscure. Nonetheless, it is widely accepted that a combination of genetic susceptibility and environmental influences contributes to the pathogenesis of this condition. The association of specific human leukocyte antigens (HLA)-DRB1*1104, -DQB1*0501, and -DQB1*0301 with SSc is well established. Other genetic loci, including PTPN22, NLRP1, STAT4, and IRF5 have also been linked to a predisposition to SSc [2]. Environmental factors, including cytomegalovirus and Epstein–Barr virus, medications, as well as pollutants like silicon dioxide, organic solvents, pesticides, and silicone, contribute to the development of SSc triggering the pathogenesis [3,6,7].

The pathogenesis of this autoimmune disorder involves a complex of innate and adaptive immune responses, characterized by early microvascular changes and endothelial cell dysfunction, followed by the transformation of endothelial cells into myofibroblasts and the production of specific autoantibodies, which play a significant role in the development of SSc. The progression of the disease leads to fibrosis and ischemia, causing irreversible damage of the affected organs and ultimately leading to failure [8]. Activated macrophages, monocytes, and dendritic cells contribute to vascular damage and fibrosis by stimulating T- and B-cells and producing cytokines that promote fibrosis and inflammation [9]. Although significant progress has been made in understanding the mechanisms behind the development of SSc, an effective treatment has yet to be found.

The presence of specific antibodies in the serum of patients with SSc influences the clinical presentation and can help making an accurate diagnosis. Specific SSc antibodies include anti-topoisomerase I (anti-topo I)/anti-Scl-70, anti-centromere antibodies (ACAs), anti-U3 ribonucleoprotein (RNP)/anti-fibrillarin, anti-U1 RNP, anti-U11/12 RNP, RNA polymerase III, and nucleolar antigens, such as anti-Th/To, anti-NOR 90, anti-Ku, and anti-RuvBL1/2, as well as anti-PM/Scl antibodies [2]. These autoantibodies serve not only as diagnostic indicators, but are also closely associated with the diverse clinical manifestations of SSc. Thus, ACAs are characteristic of the localized form of SSc (lcSSc), while the presence of antibodies to anti-topo I and RNA polymerase III indicate the diffuse cutaneous form (dcSSc) [2].

SSc can be classified into dcSSc and lcSSc, also known as CREST syndrome, which is characterized by calcinosis, Raynaud’s phenomenon, oesophageal dysfunction, sclerodactyly, and telangiectasia. LcSSc mainly affects the skin and underlying tissue, while dcSSc affects the internal organs and systems [9]. The earliest signs of SSc may include Raynaud’s phenomenon which could result in chronic finger ischemia, causing sores and gangrene [10]. Given the early occurrence of vasospastic reactions in finger vessels, this finding is an early indication that usually precedes fibrosis. This suggests that vascular disorders may be an initial manifestation of the disease and play a significant role in its pathogenesis [4]. The gastrointestinal tract is the second most affected system after the skin [5]. SSc can affect the upper and lower portions of the digestive system [5,11]. However, interstitial lung disease is a major cause of mortality among SSc patients [12]. According to the available data, approximately 8–12% of individuals with SSc develop pulmonary arterial hypertension (PAH), which is a serious complication that usually occurs 10 to 15 years after the initial diagnosis of SSc and worsens the prognosis of this disease dramatically [4].

In addition, SSc often affects the joints and muscles, although this may go unnoticed if the internal organs have been severely damaged. Due to the concurrent occurrence of skin fibrosis and subcutaneous edema, it can be difficult to determine the presence of joint effusion or the extent of joint mobility [13].

SSc-related renal crisis is characterized by the sudden onset of renal failure and severe hypertension. Recently, there has been a marked decrease in the incidence of kidney damage associated with SSc, which can be attributed to the widespread use of angiotensin-converting enzyme inhibitors becoming a standard treatment method [5].

Given the multisystemic and heterogeneous nature of systemic sclerosis (SSc), no single therapeutic approach currently provides uniform efficacy across patients. The disease often progresses to significant functional impairment, reduced quality of life, and increased mortality. This complexity underscores the need for continued investigation into its molecular mechanisms and for the development of targeted therapeutic strategies. Comparative analyses of the results of relevant global studies of genetic mutations, the spectrum of antibodies and cytokines in patients with SSc to identify commonalities, and the differences and gaps in existing knowledge are needed. The data obtained on genetic variability in various populations of SSc patients is of considerable interest, albeit no comprehensive immunogenetic studies of SSc have been conducted in Kazakhstan.

The objective was to evaluate the genetic landscape and autoantibody profile in a Kazakh cohort of patients with systemic sclerosis.

## 2. Materials and Methods

A total of 26 adult patients of Kazakh ethnicity with a confirmed diagnosis of the diffuse form of SSc according to the 2013 ACR/EULAR classification criteria were recruited for this study. Patients were recruited from the outpatient clinics and inpatient therapeutic department of the National Scientific Medical Center between August 2023 and June 2024, providing written informed consent prior to participation. All patients underwent clinical evaluations to assess disease activity and organ involvement using the European Scleroderma Study Group (EScSG) activity index and the modified Rodnan skin score (mRSS). For the control group, 18 healthy individuals of Kazakh ethnicity, with no personal or family history of autoimmune or connective tissue disease, were selected. Control participants were matched by age and sex where possible to reduce confounding. They were enrolled through routine health check programmes. All participants in both groups were assigned anonymized codes specifically for this research: individuals with SSc were labelled with codes beginning with “S” (systemic sclerosis), while control group participants were categorized as “C” (controls). Only non-identifiable data (diagnosis and sex) were collected to ensure confidentiality. No data compromising patient confidentiality was collected or included. The recruitment process was approved by the Institutional Human Ethics Committee (protocol no. 085/KI-79).

### 2.1. Ethical and Local Approvals

This project ensured compliance with the principles of scientific ethics, maintaining high standards of intellectual honesty and preventing the fabrication, falsification, and plagiarism of scientific data. This study was conducted in accordance with the Helsinki Declaration on Ethical Principles for Medical Research Involving Human Subjects. The study protocol was approved by the Institutional Human Ethics Committee of the National Scientific Medical Center (no. 085/KI-79). Written informed consent was obtained from each participant. All subjects involved in the study were informed about the purpose, objectives, and their role in the project.

### 2.2. Sample Selection and Preparation for Analysis

Blood samples were collected from the patients with SSc and healthy individuals from the control group after obtaining their informed consent. A complete blood count was performed immediately upon delivery to the main laboratory. For genetic and immune analysis, samples were aliquoted, stored, and processed with the integrity of the genetic material. In total, 26 samples from SSc patients and 18 samples of the control group were analyzed.

#### 2.2.1. Immune Markers of Systemic Sclerosis

The presence of antinuclear factor was tested using an ANA plus kit (Medipan and Generic Assays GmbH, Berlin, Germany) using an indirect immunofluorescence assay (IFA). The cell substrate was immobilized on microscope slides. In the first stage, control samples and pre-diluted patient serum at a titre of 1:80 in a volume of 25 µL were placed in the wells on the slide, followed by incubation in a humid chamber for 30 min. Then unbound immune complexes were removed by washing for 5 min twice in a Coplin jar with washing buffer. Antibodies bound to antigens in HEp-2 cells interacted with 25 µL of conjugate (fluorescein-labelled secondary polyclonal antibodies—FITCs), incubated in a humid chamber for 30 min, after which the washing stage was repeated. Next, a covering buffer (PBS buffered solution with glycerol, containing DAPI and sodium azide) was applied to each well on the microscope slide. The prepared slides were scanned using the Aklides automatic system with a description of the type of fluorescence (glow) and titre in accordance with the International Consensus on Antinuclear Antibody Patterns (ICAP, anapatterns.org).

Subsequently, specific autoantibodies, including antibodies to topoisomerase I (Scl-70), U1 small nuclear ribonucleoprotein (U1-snRNP), centromere protein B (CENP-B), the 52 kDa Ro/SSA antigen (SS-A/Ro52), the 60 kDa Ro/SSA antigen (SS-A/Ro60), the La antigen (SS-B), the Smith/ribonucleoprotein complex (Sm/RNP), the Smith antigen (Sm), ribosomal phosphoprotein P0 (Rib-P0), and nucleosome were determined by immunoblotting in accordance with the national clinical recommendations and international guidelines. Antibodies to histones, anti-dsDNA, and Jo-1 specific for systemic lupus erythematosus (SLE) and polymyositis rather than SSc were included because of the same ANA 12 Line kit (Medipan and Generic Assays GmbH, Berlin, Germany) for the qualitative determination of the autoantibodies. In the first stage, antibodies in serum samples react with antigens immobilized on the membrane of the test strip (20 µL of serum introduced into 1.5 mL of diluent). The reaction mixture with the test strip was incubated for 30 min at room temperature on a shaker at 150 rpm. For the next step, unbound immune complexes were removed by 3 washing steps using 1.5 mL of washing buffer. In the second stage, formed antigen–antibody complexes interacted with 1.5 mL of conjugate (anti-human-IgG, conjugated with horseradish peroxidase) incubated for 30 min at room temperature on a shaker at 150 rpm. After washing, the 3,3′,5,5′-tetramethylbenzidine substrate was added, which stains the corresponding antigen field on the test strip. To stop the reaction, the strips were incubated in the dark for 30 min at room temperature on a shaker at 150 rpm following by 3 min washing. The results were evaluated on a scanner using the BlotGAlaxy programme (Medipan and Generic Assays GmbH, Berlin, Germany).

The samples that tested positive for any antibodies were subjected to additional testing using the ANA cytobead kit (Medipan and Generic Assays GmbH, Berlin, Germany) to determine the quantitative concentration in IU/mL of antibodies to Smith, SS-A/60, SS-A/52, SS-B, Scl-70, CENP-B, RNP/Sm, and dsDNA. The ANA cytobead kit is a combined reagent that includes the semi-quantitative determination of antinuclear factor on HEp-2 cells and the quantitative determination of specific antibodies by the indirect immunofluorescence assay in serum. A substrate with HEp-2 cells and microparticles (beads) labelled with antigens were immobilized on a slide. For the first stage, controls, calibrators, and pre-diluted patient sera in a volume of 80 µL were placed on the slide and incubated in a humid chamber for 30 min. Then unbound immune complexes were washed off (4 times for 2 min in a Coplin jar with washing buffer). The antibodies bound to antigens in HEp-2 cells and on microparticles interacted with 80 µL of conjugate (FITC). Incubation was performed in a humid chamber for 30 min, after which the washing stage was repeated. Next, a covering buffer (PBS buffered solution with glycerol, containing DAPI and sodium azide) was applied to each well on the microscope slide and scanned using the Aklides automatic system. The results were reported with a description of the type of fluorescence (glow), titre, and antibody concentration in IU/mL.

#### 2.2.2. Cytokine Measurement

The levels of pro-inflammatory interleukin (IL)-6 in serum were quantified using an enzyme-linked immunosorbent assay (ELISA) using the Vector-Best kit on the Alisei Quality System (Calenzano, Italy). For the first stage, experimental and control samples were incubated for 60 min at 37 °C on a shaker at 700 rpm with immobilized monoclonal antibodies followed by washing. Bound IL-6 reacted with conjugate polyclonal human IL-6 antibody with biotin in a volume of 100 µL in each well during incubation for 30 min at 37 °C on the shaker. The washing was repeated. For the third step, bound conjugate no. 1 reacted with streptavidin–horseradish peroxidase in a volume of 100 µL in each well for 30 min at 37 °C on a shaker at 700 rpm. The amount of bound conjugate no. 2 was determined by a colour reaction using tetramethylbenzidine substrate in a 100 µL volume for 25 min at a room temperature of 18–26 °C in the dark. After stopping the reaction by adding a stop reagent of 100 µL, the optical density (OD) was measured by a spectrophotometer at a wavelength of 450 nm (reference wavelength 620–655 nm). The concentration of IL-6 in the analyzed serum samples was determined using a calibration curve in the range of 0.0–300.0 pg/mL.

### 2.3. Genetic Analysis

#### 2.3.1. Genomic DNA Extraction

DNA extraction from blood samples was performed using the Genejet^TM^ Whole Blood Genomic DNA Purification Minikit (Thermo Fisher Scientific, Waltham, MA, USA). The main stages included the lysis of cells to release genomic DNA and the elution of DNA by filtration, which facilitated the removal of proteins and other contaminants. Qubit fluorimetry was used together with the QUBIT 1X DSDNA HS KIT (both Thermo Fisher Scientific, Waltham, MA, USA) to perform the task of the quantitative determination of the isolated DNA.

#### 2.3.2. Preparation of a Library for Targeted Sequencing

To perform targeted sequencing of genes that undoubtedly belong to the group of autoimmune diseases, special panels were used via Autoimmune Panel, which was created specifically for this project [14].

The gene panel consists of 120 genes mainly involved in the development of SSc, but also other autoimmune conditions such as SLE and rheumatoid arthritis (RA).

#### 2.3.3. Amplification of Target Regions

Multiplex amplification was performed using the ION AMPLISEQ LIB KIT PLUS (Thermo Fisher Scientific, Waltham, MA, USA). As a result of amplification, the target DNA regions were isolated and concentrated, which ensured a high accuracy and sensitivity in the subsequent stages, including sequencing.

Each sample was barcoded using the ION DUAL BARCODE KIT (Thermo Fisher Scientific, Waltham, MA, USA), allowing for sample tracking and minimizing the risk of cross-contamination. This allowed for multiple samples to be sequenced simultaneously while maintaining sample identity.

Prepared libraries were subjected to AMPURE XP magnetic particle purification, which effectively removed unwanted contaminants and improved sample quality.

Emulsion PCR was performed using the Ion PI™ Hi-Q™ OT2 200 (Thermo Fisher Scientific, Waltham, MA, USA), which is designed for emulsifying DNA fragments. During PCR, DNA molecules are attached to microscopic carriers, allowing for the sequencing of multiple DNA fragments in parallel. Library preparation was performed on Ion PI™ v3 chips (Thermo Fisher Scientific, Waltham, MA, USA), which have a high density and allow for deep sequencing, ensuring the accurate analysis of target genetic regions.

### 2.4. High-Throughput Sequencing and Bioinformatics Analysis

High-throughput sequencing was performed using the Ion Proton platform with the Ion PI™ Hi-Q™ 200 kit to investigate the genetic landscape of SSc. Genomic DNA, extracted as described in Section 2.3.1, underwent targeted sequencing of a custom AmpliSeq panel (Thermo Fisher Scientific, Waltham, MA, USA) comprising 120 genes associated with autoimmune and fibrotic pathways. Library preparation involved multiplex amplification (Ion AmpliSeq Library Kit Plus) and sample barcoding (Ion Dual Barcode Kit) to ensure accurate sample tracking. Emulsion PCR was conducted on the Ion PI™ Hi-Q™ OT2 200 device, followed by sequencing on Ion PI™ v3 chips to achieve high-depth coverage.

#### 2.4.1. Bioinformatics Workflow

The bioinformatics pipeline utilized Ion Reporter software (version 5.20, Thermo Fisher Scientific, Waltham, MA, USA) for data processing and variant analysis. Raw sequences underwent quality control to exclude erroneous reads and short sequences, ensuring dataset reliability. Filtered sequences were aligned to a reference genome to identify polymorphisms, insertions/deletions, and structural variants. Variants were annotated using dbSNP and ClinVar databases, with filters applied to prioritize missense, nonsense, frameshift, and loss-of-function mutations. Additional parameters included a minimum read depth of 30, gene extension size to capture regulatory regions, and splice site analysis targeting intronic regions upstream of exons in a strand-dependent manner. Associations with autoimmune diseases were evaluated using locus and allele-specific annotations from ClinVar and dbSNP.

#### 2.4.2. Bioinformatic Analysis

The contemporary interpretation of genetic variants is based on a system of criteria developed by the American College of Medical Genetics and Genomics (ACMG) in collaboration with the Association for Molecular Pathology (AMP). This system provides a comprehensive evaluation of a variant’s molecular characteristics, population data, bioinformatic predictions, functional study results, and clinical correlations. The key pathogenic criteria include the following: PVS1 (null variants in genes with a known loss-of-function mechanism), PS1 (identical to a known pathogenic variant at the protein level), PS2/PM6 (de novo occurrence), PS3 (confirmed functional impact), PS4 (significant overrepresentation in a patient cohort), PM1 (location in a critical functional domain), PM2 (low population frequency), PM3 (observed in trans with a pathogenic variant), PM4 (protein length alteration), and PM5 (a different amino acid change at a position of a known pathogenic variant). Supporting criteria (PP1–PP5) encompass familial segregation data, in silico analysis results, and phenotype specificity. Benign criteria include BA1 (high population frequency), BS1-BS2 (conflicting segregation data), and BP1–BP7 (a lack of functional impact). The integration of these weighted criteria according to established combinatorial rules enables the standardized classification of variants into five categories: pathogenic, likely pathogenic, variant of uncertain significance, likely benign, and benign.

The classification of identified genetic variants was carried out using specialized expert platforms Varsome and Franklin, which integrate data from clinical and population databases, as well as results from functional studies. These resources provide preliminary automated interpretation; however, the final assignment of pathogenicity class was performed in accordance with the recommendations of the ACMG, with mandatory consideration of the clinical context.

The ACMG/AMP system is based on a standardized set of criteria with varying levels of evidence strength:

**PVS** (pathogenic very strong)—very strong evidence of pathogenicity (e.g., nonsense mutations; splice site variants; large deletions/duplications leading to a complete loss of gene function, if loss-of-function is an established disease mechanism).

**PS** (pathogenic strong)—strong evidence (e.g., variant identified de novo with confirmed parentage; reproducible functional studies showing a deleterious effect; variant observed frequently in affected individuals and absent in the general population).

**PM** (pathogenic moderate)—moderate evidence (e.g., localization in a critical functional domain; extremely low frequency in population databases; amino acid change with radical differences in biochemical properties).

**PP** (pathogenic supporting)—supporting evidence (e.g., patient phenotype consistent with the disease spectrum known for the gene; multiple in silico predictions suggesting pathogenicity).

For benign classification, the following criteria were applied:

**BA** (benign stand-alone/absolute)—stand-alone benign evidence (e.g., high population frequency incompatible with a severe inherited disorder).

**BS** (benign strong)—strong benign evidence (e.g., functional studies demonstrating no deleterious effect; occurrence in healthy adults in the homozygous state).

**BP** (benign supporting)—supporting benign evidence (e.g., in silico predictions indicating no impact; localization in a non-coding region without effect on splicing).

The combination of these criteria allowed the classification of variants into one of five categories: pathogenic, likely pathogenic, variant of uncertain significance (VUS), likely benign, or benign. This system ensures the reproducibility of variant interpretation and the comparability of results across laboratories.

As an example, the variant SAMD9L chr7:92761606 GT/G can be considered. According to Varsome and Franklin, this variant is located in the SAMD9L gene, which is associated with a predisposition to myelodysplastic syndromes and autosomal dominant bone marrow failure syndromes. Interpretation according to ACMG criteria included an extremely low frequency in population databases (PM2), localization in a functionally important region (PM1), and multiple in silico predictions supporting pathogenicity (PP3). Based on these data, the variant was classified as likely pathogenic.

#### 2.4.3. Comparative Population Frequency Databases and Variant Annotation Resources

To estimate the frequency of detected variants in various populations, several international genomic databases were consulted, representing aggregated sequencing data from thousands to hundreds of thousands of individuals of diverse ethnic and geographic origins.

The Genome Aggregation Database (gnomAD) serves as the most comprehensive resource of aggregated human genetic variation, including exome and genome sequences from more than 140,000 individuals (version v2.1.1) and over 760,000 individuals in version v3.1.2. Its primary purpose is to estimate allele frequencies and to facilitate the distinction between common polymorphisms and rare pathogenic variants relevant to both Mendelian and complex diseases [15].

The 1000 Genomes Project remains a foundational dataset in population genetics, encompassing 2504 individuals from 26 global populations representing five super-populations (African, East Asian, European, South Asian, and American). This resource provides a reference spectrum for common genetic variation (>1% minor allele frequency), making it valuable for assessing allele prevalence in diverse ancestries [15].

The Database of Single Nucleotide Polymorphisms (dbSNP), maintained by the National Center for Biotechnology Information (NCBI), serves as a central repository of SNPs and other short genetic variants. Each variant is assigned a reference SNP ID (rsID), enabling harmonized reporting and cross-database integration. This database is essential for variant annotation, the identification of known polymorphisms, and interoperability with other genomic resources [16].

The Trans-Omics for Precision Medicine (TOPMed) programme, established by the National Heart, Lung, and Blood Institute (NHLBI), provides deep whole-genome sequencing data for more than 180,000 individuals. TOPMed’s emphasis on ancestral diversity enhances its capacity to reveal rare variants relevant to cardiovascular, pulmonary, and hematologic diseases [17].

### 2.5. Statistical Analysis

The statistical analyses were performed using the GraphPad Prism 8 and IBM SPSS version 21.0. Hierarchical clustering and heatmap visualization were performed using Ion Reporter 5.20 software (ThermoFisher Scientific, Waltham, MA, USA). The statistical evaluation of categorical variables was conducted using Fisher’s exact test and the Chi-square (χ^2^) test, depending on the distribution and expected frequencies within the contingency tables. A comparative analysis using the χ^2^ test with Yates’s correction was performed to evaluate the relationship between specific autoantibodies/genetic variations and clinical manifestations in patients with SSc. Statistical comparisons of IL-6 levels in patient and control groups were compared with the Mann–Whitney test since data was determined to not follow a normal distribution, while for data on antibodies the Wilcoxon signed-rank test was performed. A *p*-value of <0.05 was considered statistically significant.

## 3. Results

The main group consists of 26 Kazakh patients who have had SSc for the last 9.65 ± 9.8 (mean ± SD) years. mRSS was equal to 16 ± 7.2 and EScSG to 3.54 ± 2.18. In the majority of patients, 96% (25/26), various skin lesions were observed, including skin thickening, varying in extent, dense edema, “purse-string” mouth, sclerodactyly, digital ulcers, puffy fingers, and hyper- and hypopigmentation (Figure 1). Articular injury manifested in the form of polyarthralgia or arthritis was found in 23/26 patients (88%), as well as vascular disturbances (Raynaud’s phenomenon). Overall, 19 of 26 patients had frequent lesions of the gastrointestinal tract, in the form of esophagitis (73%). Seventeen patients suffered from lung impairment, which manifests as interstitial lung disease (SSc-ILD) with signs of non-specific interstitial pneumonia (NSIP, more common) or usual interstitial pneumonia (UIP) that has been found on high-resolution computed tomography (Figure 2). Ten patients (38%) were diagnosed with a secondary Sjögren’s syndrome, characterized by xerostomia and xerophthalmia. Three patients had thyroid impairment (12%) and two cardiovascular disorders (7.6%, Appendix A).

According to the obtained treatment, 11 out of 26 patients (42%) were receiving Methylprednisolone with an average daily dose of 5.36 ± 2.11 mg/day. Five patients have been treated with Mycophenolate mofetil, in combination with Methotrexate (one patient) and with Leflunomide (one individual). D-penicillamine was prescribed to 5 patients, in combination with Methotrexate for 1 patient and Leflunomide for 1 individual. A single patient received Azathioprine as a monotherapy. Leflunomide alone was given to five patients with prominent polyarticular syndrome, three patients obtained Methotrexate, and one patient was treated with Plaquenil. Antifibrotic therapy (Nintedanib) was added to the treatment strategy for two patients.

### 3.1. Antibodies Spectrum and Pro-Inflammatory Profile in SSc Patients

The mean titre of antinuclear factor (ANF) in the study group was equal to 1255.7 ± 1596.25. The antinuclear factor on HEp-2 cells was positive (at a titer > 1:160) in the majority of patients, most commonly at 1:640 (38.5%), while negative (at a titer < 1:80) in only 3/26 patients (11.5%, Table 1). In contrast, the control group showed exclusively negative ANF results (18/18, 100%). The difference between groups were statistically significant (*p* < 0.0001, Figure 3).

The presence of antinuclear antibodies was detected in varying concentrations among patients with different types of glows (Figure 3). The main types of HEp-2 cell glow observed in the group of SSc patients were granular (AC-4/5, antibodies to SS-A/Ro60 and SS-A/Ro52) and centromere (AC-3, antibodies to CENP-B, Figure 3). Different variations in the cytoplasmic type of glow (AC-19, AC-20, and AC-21) were also found, indicating the presence of antibodies such as SRP, RNP, and ribP0 (Table 2, Figure 3).

Patients in the SSc group demonstrated variable antibodies, the level of which were undetectable in the control group of healthy individuals (Appendix A). In the main group of SSc patients, antibody data analysis showed antibodies to CENP-B and Scl-70 which are diagnostic criteria for SSc (*p* < 0.0001, Figure 4). Anti-CENP-B was found in 38.5% of the examined patients. High anti-Scl-70 was detected in a male patient, with a rapidly progressive course of SSc, ILD, pulmonary arterial hypertension, Raynaud’s syndrome, skin lesions (dense edema, hyperpigmentation, and induration), esophagitis, and arthritis. This patient was also diagnosed with typical secondary Sjogren’s syndrome (sSS) with sialoadenitis and xerostomia, associated with anti-SS-A/60 and SS-A/52 positivity.

Antibodies to SS-A/Ro60 and SS-A/Ro52 were significantly higher in the main group (*p* < 0.001, Figure 4). In particular, SSc patients who had antibodies to SS-A/Ro60 and SS-A/Ro52 demonstrated Sjögren’s syndrome (sialoadenitis, xerostomia). Anti-U1-snRNP was found in 23.1% of cases as well as RNP/Sm. Two patients demonstrated positivity for anti-rib-P0 (7.7%).

Antibodies to nucleosomes, histones, and anti-dsDNA (which are more specific for SLE than SSc) were positive in isolated cases of the main group (Table 3, Figure 5C).

A comparative analysis using the χ^2^ test with Yates’s correction was performed to evaluate the relationship between specific autoantibodies and distinct clinical features in patients with SSc. High χ^2^ values and statistically significant *p*-levels (*p* < 0.05) were observed for anti-Scl-70, anti-SS-B, anti-nucleosome, anti-Sm, anti-U1-snRNP, anti-RNP/Sm, and anti-ribosomal P0 antibodies in relation to cutaneous, joint, vascular, pulmonary, and esophageal involvement. Among these, anti-Scl-70, anti-CENP-B, and anti-nucleosome antibodies showed the strongest association with diffuse skin fibrosis, vascular, and joint involvement (χ^2^ = 37–44, *p* < 0.0001).

In our cohort, antibodies to SS-A/Ro52 and SS-A/Ro60 demonstrated significant associations with cutaneous, joint, and vascular involvement according to Yates-corrected χ^2^ analysis (*p* < 0.0001). The SS-A/Ro60 antibody additionally showed a weaker but statistically significant association with pulmonary manifestations and esophageal involvement (*p* < 0.02), suggesting a broader spectrum of organ-related activity. In contrast, Ro52 reactivity also demonstrated relevance to cardiac and hepatic (cirrhotic) complications (*p* = 0.0143 and *p* = 0.0049, respectively), highlighting its potential contribution to systemic tissue damage and fibrotic processes beyond the skin.

Antibodies to CENP-B correlate with vasculopathy and skin lesions with an elevated modified Rodnan skin score (*p* < 0.005). They demonstrated a distinct pattern, correlating primarily with cardiac and thyroid involvement (*p* < 0.003), which is consistent with limited cutaneous forms of SSc.

A moderate association was detected between several antibody clusters (anti-Scl-70, nucleosome, SS-B, and histone) and secondary Sjögren’s syndrome (*p* = 0.006). Overall, these findings confirm the strong immunoclinical characteristic of SSc, reflecting both the systemic nature of the disease and the diagnostic value of extended autoantibody panels in stratifying organ-specific risk.

In the control group, IL-6 levels were normal in all participants, suggesting stable baseline cytokine activity in healthy individuals. In the main group, although the majority also fell within the normal range (88.5%, 23/26), a subset (11.5%, 3/26) demonstrated elevated IL-6. The difference between groups was statistically significant (*p* = 0.014, Figure 4). Statistical analysis using χ^2^ with Yates’s correction revealed a significant association between a high level of IL-6 and skin lesions, joint impairment, and Raynaud’s phenomenon (χ^2^ = 30–38, *p* < 0.0001). Elevated IL-6 was also associated with pulmonary involvement and esophageal pathology (χ^2^ = 13.7 and χ^2^ = 17.7, respectively, *p* < 0.0001).

### 3.2. Genetic Variability

Genetic analysis revealed multiple pathogenic variants of 14 genes, including SAMD9L, Ly96, REL, IRAK1, RBPJ, IL6ST, TNFAIP3, ITGA2B, ABCC2, AIRE, IL6R, JAZF1, IKZF3, and AFF3, in the group of SSc patients (Table 4). According to Fisher’s exact test, no significant differences (*p* < 0.05) were observed between SSc and control groups. A slight, though not statistically significant, trend was noted for the LY96 gene variants as indicated by the two-tailed Fisher’s exact test (Table 4). This subtle variation may reflect the limited statistical power resulting from the small sample size. Nonetheless, this gene merits attention in future studies with larger cohorts, where increased statistical power may uncover meaningful differences in allele frequencies between patient and control groups. Extended gene variant annotations, types, ACMG classifications, as well as population data can be found in Appendix A. The TOPMed, gnomAD, and dbSNP international genomic databases were used to estimate the frequency of detected variants from diverse ethnic and geographic origins [15,16].

For SAMD9L, we detected two distinct variants that were classified as likely pathogenic according to ACMG criteria in eight individuals (representing 18.2% of the screened cohort). Previous research demonstrated that SAMD9L mutations correlated with cytopenia, and inflammatory presentations were previously published [18,19]. Our results revealed that four of these eight carriers also presented with autoimmune features. Notably, two patients exhibited the compound heterozygous state for the SAMD9L findings, mirroring earlier observations of functional loss-of-function or gain-of-function changes that can lead to complex autoinflammatory phenotypes [20].

In the main group, eight patients have variants in the SAMD9L gene (chr7:92764981 T/TT; chr7:92761606 GT/G). Both variants are likely pathogenic according to the ACMG classification. In our study, half of the SSc patients with likely pathogenic variants in the SAMD9L gene (four out of eight) suffered from Sjogren’s syndrome, while two had a compound heterozygous state. However, no significance has been found using the Yates-corrected Chi-square test.

A likely pathogenic variant in the REL gene (chr2:61149099 GT/G) was identified in five patients with SSc and in two individuals from the control group, suggesting a potential enrichment in the affected cohort. This variant exhibited significant associations with articular, cutaneous, and vascular pathologies (χ^2^ = 27.9–34.7, *p* < 0.0001). In addition, three distinct variants in the RBPJ gene were observed exclusively among three SSc patients, whereas no such variants were detected in the control group. All these RBPJ variants were classified as VUS.

The chr6:138199775 T/TC substitution in the TNFAIP3 gene, classified as likely pathogenic, was identified in two SSc patients and associated with cutaneous, vascular, and joint manifestation, as well as ILD and esophagitis (χ^2^ = 16.3–40.9, *p* < 0.0001).

Four patients from the main group carried variants in the ABCC2 gene. Two individuals harboured a pathogenic variant (chr10:101578956 CA/C), while two others carried likely pathogenic variants (chr10:101603641 CA/C and chr10:101559041 CA/C). None of these variants were detected among control subjects. Notably, all four SSc patients with ABCC2 gene alterations exhibited a vascular phenotype, including Raynaud’s phenomenon and other signs of vascular injury (χ^2^ = 27.9, *p* < 0.0001), suggesting a possible contribution of ABCC2 dysfunction to endothelial or microvascular pathology.

Variants in the IL6R gene were identified in two patients with SSc but were absent in the control group. The variant chr1:154378136 GC/G was classified as likely pathogenic, whereas chr1:154401686 G/A was defined as a VUS.

Additionally, two SSc patients carried the chr17:37922552 T/C variant in the IKZF3 gene, also classified as a VUS, and associated with cutaneous, articular, and vascular impairment, as well as ILD and esophagitis (χ^2^ > 16, *p* < 0.0001).

Overall, 10 rare or low-frequency variants were identified across six genes within the SSc cohort. Among these, 3 variants met the criteria for likely pathogenic or pathogenic, while 25 variants were classified as VUS. A gene-by-gene summary is presented in Appendix A.

A heterozygous variant was detected in the LY96 gene. Approximately 42% of carriers exhibited elevated ESR levels, whereas IL-6 concentrations remained within the normal range. This pattern may reflect the functional role of LY96 (also known as MD-2) as a co-receptor in toll-like receptor (TLR) signalling, modulating innate immune activation. These findings are consistent with previous reports suggesting that LY96 polymorphisms influence immune responsiveness in rheumatic diseases [21]. LY96 and PTPN22 variants were associated with an elevated Rodnan skin score (χ^2^ = 10.3–19.1, *p* < 0.001). Although most variants showed no significant association with thyroid, liver, or cardiac involvement, two genes—LY96 and PTPN22—reached marginal significance for thyroid, hepatic, and cardiac pathologies (*p* = 0.01).

Three distinct variants were found in the IRAK1 gene, representing about 8% of the total cohort, making it one of the more frequently altered genes in this study. Notably, two of these variants were in proximity to the previously reported 196Phe/532Ser functional haplotype associated with SSc subsets [22]. Patients harbouring IRAK1 variants exhibited more severe clinical manifestations, including diffuse cutaneous lesions and interstitial lung disease, consistent with earlier findings linking IRAK1 polymorphisms to enhanced disease activity and fibrosis pathways in SSc [22,23].

A heterozygous missense variant was detected in the ITGA2B gene. The affected patient demonstrated elevated platelet activation indices alongside increased fibrosis markers, suggesting the potential involvement of platelet–fibroblast crosstalk in vascular remodelling. However, given the limited sample size, further studies are required to determine whether ITGA2B alterations contribute significantly to disease severity or vascular alterations.

A single likely pathogenic variant in AIRE was discovered. Clinically, the individual harbouring this change presented with a mixed overlap phenotype, including autoimmune thyroiditis. Given the well-established role of AIRE in maintaining central immune tolerance, even heterozygous changes may predispose patients to systemic autoimmunity by impairing the thymic selection of autoreactive lymphocytes. However, validating this hypothesis will require further studies.

A previously undescribed variant was detected in the JAZF1 gene, known for its roles in metabolic regulation and, more recently, immune modulation. This variant was found in two patients, one of whom—a 62-year-old woman—had a chronic course of SSc with cutaneous involvement, esophagitis, and secondary Sjögren’s syndrome (sialadenitis, xerostomia) for more than two years. Following a SARS-CoV-2 infection, the patient noticed dry skin, weight loss, and salivary gland inflammation. On examination, a slight lump on the fingers was noted, and esophagitis was confirmed endoscopically. The patient was treated with Mycophenolate mofetil (1500 mg/day) and Methylprednisolone (8 mg/day) with gradual tapering. The second patient carrying the JAZF1 variant was a 61-year-old man with subacute SSc, diffuse cutaneous form, with cutaneous, vascular, and pulmonary involvement and esophagitis. The disease onset occurred at 11 years of age, marked by the first manifestations of Raynaud’s phenomenon. Clinically, he presented with hyperpigmentation, dense edema, and induration of the hands, as well as moderate thickening of the facial and pedal skin (mRSS = 16). Additional features included digital ulcers, sclerodactyly, arthralgia, Raynaud’s vasospasm, and SSc-ILD with stage I respiratory insufficiency. Gastrointestinal evaluation revealed esophagitis and gastritis. The patient received Methylprednisolone (4 mg/day) and D-penicillamine (250 mg/day). Notably, cutaneous fibrosis was a shared phenotype among both JAZF1 variant carriers (χ^2^ = 40.9, *p* < 0.0001), suggesting a potential link between JAZF1-related pathways and fibrotic remodelling. The association was also found with skin lesions, Raynaud’s phenomenon, ILD, and esophagitis (χ^2^ = 16.2–40.9, *p* < 0.0001), though further studies with larger cohorts are required to determine whether JAZF1 contributes synergistically to pro-fibrotic and vascular mechanisms in SSc.

Finally, two likely pathogenic variants were identified in AFF3, a transcription factor that can impact lymphocyte development. The first carrier, a 25-year-old woman, was diagnosed with chronic SSc, with mild disease activity and multisystem involvement. Her disease course, spanning 11 years, began with classical Raynaud’s phenomenon. Clinically, she exhibited vascular manifestations, gastrointestinal dysfunction, arthralgia, and cutaneous fibrosis characterized by facial edema, perioral tightening (“purse-string” mouth), and skin thickening on fingers and feet, with lesser involvement of the forearms and shins (mRSS = 18). The patient also had secondary Sjögren’s syndrome and was treated with D-penicillamine (500 mg/day). A Yates-corrected Chi-square test revealed the association between AFF3 variants and cutaneous, articular, and vascular impairment, ILD, and esophagitis (χ^2^ = 19.1–44.8, *p* < 0.0001); a weak association was found with Sjögren’s syndrome (χ^2^ = 7.4, *p* < 0.001).

Overall, the analysis of immunogenetic variants and specific organ manifestations revealed multiple statistically significant genotype–phenotype associations in SSc. Although many of the identified variants occurred at relatively low frequencies, their collective pattern underscores the polygenic and immunogenetically complex nature of SSc. Genes related to innate immune signalling and cytokine pathways—PTPRC, CTLA4, IL6ST, SLC5A11, REL, RBPJ, CLEC16A, ITGA2B, ABCC2, AIRE, TNFAIP3, IL6R, JAZF1, IKZF3, AFF3, TREX1, IL18, IL12B, PRKCQ, PXK, DNASE1L3, and SFTPD—showed the strongest associations with skin and vascular lesions (*p* < 0.001). Variants in TNFAIP3, IL6R, JAZF1, PTPRC, CTLA4, IKZF3, AFF3, TREX1, IL18, IL12B, PRKCQ, PXK, DNASE1L3, and SFTPD exhibited significant associations with joint pathology (χ^2^ > 30, *p* < 0.0001).

ILD was associated with AFF3, TREX1, IL18, IL12B, PRKCQ, PXK, and DNASE1L3 (χ^2^ = 19.1, *p* < 0.0001). The presence of SAMD9L and IRAK1 variants in patients with SSc-ILD (χ^2^ = 4.9–7.8, *p* < 0.02) supports their potential involvement in fibrogenic signalling and interferon-mediated inflammation. A subset of variants—TNFAIP3, IL6R, JAZF1, IKZF3, AFF3, TREX1, IL18, IL12B, PRKCQ, PXK, DNASE1L3, and SFTPD—were weakly but consistently associated with Sjögren-like features (*p* < 0.02). These genes are enriched in pathways related to B-cell activation, IL-6 response, and autoantibody production, suggesting partial overlap with sSS pathogenesis.

Alterations in key immune signalling may act synergistically to shape disease susceptibility, organ-specific manifestations, and clinical heterogeneity within the Kazakh SSc cohort. Figure 6 demonstrates differential mutation profiles between the control and SSc cohorts. It should be noted that genes PTPN22, CTLA4, and SLC5A11 occurred with the same frequency in the control group and the main group. Collectively, these heatmaps illustrate that SSc samples harbour a broader and more frequent spectrum of protein-altering variants—particularly missense and nonsense—in a specific cluster of genes, compared with controls. Splice variants also appear more consistently in SSc for certain genes. These findings point to potential molecular underpinnings of SSc, such as dysregulated immune pathways or excessive extracellular matrix remodelling.

## 4. Discussion

The data obtained from 26 Kazakh individuals with SSc revealed 88.5% of antinuclear factor positivity with the presence of various antibodies in serum, which were comparative to previous research [24]. The typical SSc antibodies include anti-topoisomerase antibody and anti-centromere antibodies [25].

Anticentromere antibodies (ACAs) target the six centromere-associated nucleoproteins, designated CENP-A through CENP-F, among which peptide B (CENP-B)—a DNA-binding protein—is recognized as a major autoantigen, reacting with nearly all sera positive for ACAs. Antibodies to CENP-B were found in 38.5% of patients in our study. On average, the detection rate of antibodies to centromeres in patients with SSc is about 30% [26]. In other systemic rheumatic diseases, these autoantibodies are practically not found, apart from Sjögren’s disease and primary biliary cirrhosis. In the indirect immunofluorescence (IIF) reaction of HEp-2 cells, these autoantibodies have a typical glow, which is classified as AC-3 according to the ICAP nomenclature (centromere type of glow with separate points (40–80/cell) in the nuclei of interphase cells and in the chromatin region). The specificity of ACA detection is estimated to be between 97% and 99%, with the typical manifestations of SSc with Raynaud’s phenomenon. An association has been noted between the presence of antibodies to centromeres and the carriage of HLA-DR1, -DR4, -DR8, -DR11, and -DQ7 (DQB1*0301).

Antibodies to Scl-70 (70 kDa antisystemic sclerosis antibodies, topoisomerase I antibodies) were found only in 3.8% of the Kazakh cohort of SSc patients. Usually, they appear more often in the diffuse (40%) and less often in the limited (20%) form of SSc, CREST syndrome. They are highly specific to SSc and constitute a poor prognostic sign regarding the development of pulmonary fibrosis. Anti-Scl-70 antibodies are suggested as a predictor for a faster decline in forced vital capacity in patients with SSc-ILD [27]. Indeed, in our research, an increased level of Anti-Scl-70 was detected in one male patient, with a rapidly progressive course of SSc-ILD, pulmonary arterial hypertension, Raynaud’s syndrome, skin injury, esophagitis, arthritis, and Sjogren’s syndrome with anti-SS-A/60 and SS-A/52 antibodies.

Antibodies to SS-A/Ro are directed against proteins associated with RNA Y1–Y5 in the spliceosome. These antibodies are most often found in Sjögren’s syndrome, SSc, and SLE [28]. In SLE, their production is associated with a certain set of clinical manifestations and laboratory abnormalities: photosensitivity, Sjögren’s syndrome, and the hyperproduction of rheumatoid factor. In our study, anti-SS-A/Ro52 and SS-A/Ro60 were found in 46.2% and 30.7% of patients, respectively, and demonstrated significant associations with cutaneous, joint, and vascular involvement. A weaker but significant association of SS-A/Ro60 was detected with pulmonary manifestations and esophagitis. Anti-Ro52 demonstrated relevance to cardiac and hepatic (cirrhotic) complications which align with previous reports linking Ro52 with diffuse inflammatory responses. Together, the significant correlations observed across multiple organ domains indicate that SS-A/Ro52 and SS-A/Ro60 may serve as markers of widespread immune activation in Kazakh SSc patients rather than being restricted to a single clinical subset.

Anti-RNP are antibodies to the protein components of the small nuclear nucleotide, U1-RNP, which has been found in 23.1% of cases according to the obtained results. Anti-U1-RNP is less frequently in SSc than anti-topoisomerase, anti-centromere, or the anti-RNA-polymerase III antibody [25]. They could be found also in mixed connective tissue diseases, SLE, and other rheumatic diseases. In the IIF reaction on HEp-2 cells, autoantibodies to Sm, RNP, SS-A/Ro, and SS-B/La have a typical glow, which is classified as AC-4/5 according to the ICAP nomenclature. They were associated with cutaneous, joint, and vascular manifestations, as well as ILD and esophagitis. Recent research data described an anti-U1RNP association with ILD with a higher risk of progression and muscle and kidney lesions in SSc [29].

Antibodies to dsDNA, nucleosome, and histone were found in one patient for each antibody (3.8%). They are more specific for SLE than SSc and other autoimmune pathologies. Anti-dsDNA was included in SLE criteria and could be used for the prediction of the development of SLE before the onset of a full-blown picture of the disease [30]. Antibodies to nucleosomes are noted in 40–70% of patients with classic lupus erythematosus with skin and internal organ damage and in 80% of SLE patients with kidney impairment [13]. Rarely, they are found in patients with Sjögren’s syndrome (5–8%), and with the same frequency in mixed connective tissue disease. Antibodies to histones are detected in 80% of SLE cases, in low titres in rheumatoid or juvenile arthritis, primary biliary cholangitis, Epstein–Barr virus infection, sensory nephropathies, monoclonal gammopathies, and malignant neoplasms [31]. In the IIF reaction on HEp-2 cells, these autoantibodies have a typical glow, which is classified as AC-1 according to the ICAP.

Antibodies to Sm (Smith), directed against U1-, U2-, and U4-ribonucleoproteins, are considered highly specific for SLE and are only rarely observed in other systemic connective tissue diseases [32]. In our cohort, anti-Sm antibodies were detected in 7.7% of patients. Additionally, one patient (3.8%) demonstrated the presence of anti-SS-B/La antibodies, a protein complexed with RNA polymerase III. Such antibodies are typically associated with Sjögren’s syndrome and could be observed during the early stages of SLE.

Two patients demonstrated positivity for antibodies to rib-P0 (7.7%) which interact with ribosomal phosphoproteins and were found mainly in SLE patients. In the IIF reaction on HEp-2 cells, autoantibodies to Sm, RNP, SS-A/Ro, and SS-B/La have a typical glow, which is classified as AC-19 according to the ICAP (cytoplasmic dense finely granular). Although anti-Sm, anti-SS-B/La, and anti-rib-P0 antibodies are classically associated with SLE and Sjögren’s syndrome, their occasional presence in SSc has been documented in several cohorts [33]. This finding most likely reflects overlapping immunopathogenic mechanisms among systemic autoimmune diseases. Epitope spreading secondary to chronic inflammation and fibrosis in SSc may expose nuclear antigens, leading to secondary immune responses against ribonucleoprotein complexes such as Sm and SS-B/La. Furthermore, shared genetic determinants—particularly in loci such as IRF5, STAT4, and TNFAIP3—support the notion of a common immunogenetic background predisposing to autoreactivity across diseases. Therefore, the detection of these antibodies in a minority of SSc patients in our cohort may indicate a subclinical overlap with lupus or Sjögren’s features rather than disease-specific autoimmunity, underscoring the heterogeneity of SSc and its intersection with other connective tissue disorders [34].

A combination of antibodies was found in 46% of examined SSc patients (12 out of 26, Appendix A). In data from the literature, the prevalence of double autoantibody ranges from 0.6 to 35% [25]. The analysis of autoantibodies associated with SSc is essential for categorizing patients based on their phenotypes, vulnerability to developing organ-specific complications, and ultimately determining their overall prognosis.

The presence of autoantibodies reflects chronic B-cell activation and immune complex formation, which can engage endosomal TLR, leading to the activation of the MyD88–IRAK1–NF-κB axis. This pathway converges on the transcriptional upregulation of pro-inflammatory cytokines, notably IL-6. The identification of variants in IL6ST, IL6R, REL, and TNFAIP3 in our cohort further supports the contribution of the IL-6 signalling pathway to disease activity and fibrotic progression in SSc. An elevated level of IL-6 was detected in 11.5% of Kazakh patients with SSc which is consistent with immune activation and systemic inflammation. The finding that only a minority of patients exceeded the reference range may indicate heterogeneity in disease activity. The absence of elevated IL-6 in controls and its presence in a subset of patients highlights IL-6 as a potential marker of disease activity, even if not universally elevated. Herein we used it as a marker of acute systemic inflammation; it was correlated with a high level of antinuclear factor, followed by Raynaud’s phenomenon, skin and joint impairment, as well as the development of ILD and esophagitis (*p* < 0.0001).

Integrating serological and genetic findings provides a more comprehensive understanding of how immune dysregulation manifests in SSc patients, as well as their relevance to the Kazakh population. Genetic markers play a critical role in identifying individuals at risk for developing SSc, a condition characterized by its clinical heterogeneity. Through the examination of a diverse array of genetic variations, this study has identified several loci that appear to be particularly linked to susceptibility to SSc in the Kazakh population. Novel VUS at the following genomic coordinates were identified and have not been previously reported in association with SSc: LY96 (chr8:74922341 CT/C), PTPN22 (chr1:114381166 CT/C), IRAK1 (indels at chrX:153278833), and SAMD9L (chr7:92761606 GT/G; chr7:92764981 T/TT).

The SAMD9L gene encodes a protein that plays a key role in controlling cell growth, division, and differentiation, particularly the hematopoietic stem cells of bone marrow. Research on undiagnosed systemic autoinflammatory conditions identified six spontaneous (de novo) mutations in this gene among patients presenting with panniculitis and worsening cytopenia [18]. A previous study identified a significant association between SAMD9L expression and immune cell infiltration, suggesting its potential role as a biomarker in Sjögren’s syndrome [19]. Additionally, germline loss-of-function mutations in SAMD9L have been observed in pediatric patients with systemic autoinflammatory symptoms, though these cases did not show an increased risk for myelodysplastic syndrome or acute myeloid leukemia [20].

Variants in PTPN22, CTLA4, and REL occupy an important place within the regulatory network that maintains immune tolerance and orchestrates inflammatory balance. The PTPN22 gene encodes the lymphoid phosphatase Lyp, which dampens early T-cell receptor signalling [35]. A well-known amino acid substitution (R620W) disrupts the Lyp–CSK interaction and alters the threshold of T-cell activation [35]. Instead of ensuring efficient clonal deletion, this change favours the persistence of autoreactive lymphocytes and amplifies downstream cytokine production, particularly IL-6 and IFN-γ. Both cytokines engage the STAT3 and TGF-β pathways, promoting fibroblast proliferation and extracellular matrix synthesis that drive the fibrotic component of SSc.

The CTLA4 gene acts at a later stage of the same immune cascade, encoding an inhibitory receptor expressed on activated and regulatory T-cells. Polymorphisms such as +49A/G and CT60 have been shown to reduce receptor expression or disturb its recycling dynamics, weakening the co-inhibitory signal required to terminate antigen-driven responses [36]. The loss of this negative feedback leads to prolonged T-cell activation, excessive B-cell stimulation, and enhanced autoantibody formation. This mechanism may account for the wide spectrum of antinuclear and extractable antibodies detected in the examined cohort and supports the hypothesis that CTLA-4 dysfunction contributes to chronic immune activation in SSc.

The REL gene encodes the c-Rel subunit of the NF-κB transcription complex, which regulates a broad set of cytokines including IL-2, TNF-α, and IL-6. Variants affecting REL activity influence the differentiation of Th1 and Th17 subsets and modify the balance between inflammatory and regulatory signalling. Individuals with early SSc demonstrated reduced transcriptional activity of both c-Rel and its inhibitor, IκBα. Additionally, in cases of limited SSc, there is decreased transcriptional activity of genes encoding c-Rel and p50, subunits of NF-κB. These findings suggest that in the early stages of SSc, the NF-κB signalling pathway may be modulated by downregulating IκBα, potentially contributing to tissue homeostasis [37]. Disruptions in NF-κB signalling may impair immune system tolerance, triggering autoimmune responses. In SSc, disturbed NF-κB dynamics have been associated with persistent fibroblast stimulation and the accumulation of collagen in affected tissues [37]. The variant observed in the Kazakh cohort may reflect the altered transcriptional control of pro-inflammatory mediators consistent with the IL-6 cytokine profile.

Together, PTPN22, CTLA4, and REL represent successive checkpoints along a shared autoimmune pathway: the modulation of TCR activation, the control of costimulatory feedback, and the transcriptional regulation of effector responses. Their combined disturbance sustains lymphocyte activation, weakens tolerance mechanisms, and fosters continuous crosstalk between immune and stromal compartments. This convergence links inherited susceptibility with the clinical manifestations of fibrosis and vascular injury that define SSc.

Another important gene, the RBPJ, participates in NOTCH-dependent cellular signalling pathways, which are essential for proper cell development. In studies involving the conditional knockout of the gene’s DNA-binding domain in murine bone marrow, researchers observed its vital function in T-cell maturation. Notably, the absence of RBPJ resulted in abnormal B-cell expansion within the thymus, highlighting its involvement in lineage commitment. These findings support the idea that NOTCH-RBPJ interactions determine whether hematopoietic precursors differentiate into T- or B-cell lineages [38]. Wang et al., (2023) identified a role for the NOTCH signalling pathway as a contributing factor in the development of SSc [39].

The TNFAIP3 gene, identified by genome-wide association studies (GWAS) in patients with SSc, includes the TNF-α-induced protein 3 (TNFAIP3), also known as the A20 protein. A20 acts as a negative regulator of the TNF-induced NF-κB signalling pathway [40]. In this regard, the downregulation of A20 expression by siRNA in human foreskin fibroblasts led to a marked increase in collagen upregulation and α-smooth muscle actin (α-SMA) gene expression following transforming growth factor-β (TGF-β) stimulation, indicating that reduced A20 function may enhance TGF-β-driven fibrotic responses with collagen production [41]. Additionally, TNFAIP3-interacting protein 1 (TNIP1), which regulates TNFAIP3 activity, was also identified as a gene associated with SSc in the European population, with its association confirmed by meta-GWAS analysis [34]. In the current research, the likely pathogenic variant in the TNFAIP3 gene was associated with cutaneous, vascular, and joint manifestation, as well as ILD and esophagitis. This data was compatible to a previous study that suggested an association of TNFAIP3 with a severe form of SSc [2].

Although no direct link has been established between ABCC2 and the onset of SSc, existing research points to its involvement in autophagy and mitophagy processes [42]. Another study describes the role of these processes in the development of atherogenesis, and, in general, the impact of altered autophagy on cardiac and vascular homeostasis [43]. Notably, in a group of four Kazakh SSc patients carrying altered ABCC2 gene variants, all exhibited clinical signs of vascular involvement, including vascular injury and Raynaud’s phenomenon.

Recent findings have highlighted the involvement of genes regulating innate immune responses particularly IRF4, IRF5, IRF7, and IRF8 in SSc. These genes, previously linked to SLE, appear to contribute to disease susceptibility through overlapping immunogenic pathways [34,44]. We draw attention to the IRF5 gene which mediates the induction of pro-inflammatory cytokines such as IL6 and IL12. The IL6R gene is involved in the formation of an immune response associated with IL6 which plays a significant role in SSc development. Apart from three patients with an elevated level of IL-6, two had genetic variants of IL6: IL6R (chr1:154401686 G/A)-LP and IL6ST (chr5:55247857 GT/G)-LP. Variants in IL6R and IL6ST—key components of the IL-6 receptor complex observed in both European and East Asian studies (Appendix A)—modulate IL-6 trans-signalling, influencing vascular and pulmonary manifestations of SSc. According to our results their clinical picture included ILD and skin, vascular, and joint impairment (*p* < 0.001). The protein produced by this gene forms part of the receptor system for IL-6 which is a multifunctional signalling molecule that controls cellular development processes and has significant effects on immune system function. The complete IL-6 receptor requires both this protein component and another subunit (IL6ST/GP130/IL6-beta), which serves as a common signalling element for several different cytokines. Abnormal levels of IL-6 and its receptor components have been associated with various pathological conditions, including certain blood cancers, autoimmune disorders, and prostate malignancies [45].

The IKZF3 gene encodes a transcription factor essential for haematopoiesis and plays a key regulatory role in the maturation and function of B lymphocytes. A comprehensive analysis of 697 Chinese SSc patients identified IKZF3 variations associated with SSc [46]. Autoimmune diseases, including SSc and SLE, are characterized by a violation of self-tolerance, the activation of B-cells, followed by the elevated production of a spectrum of autoantibodies and immune complexes. Given their central role in the humoral immune response, B-cells are considered key players in the pathogenesis of autoimmune conditions and certain B-cell malignancies [47]. Notably, IKZF3 and AFF3, transcriptional regulators of lymphocyte differentiation, have been recognized in European and Middle Eastern cohorts, reinforcing their cross-population relevance to autoimmunity and fibrosis (Appendix A).

Our data support a polygenic model of organ-specific autoimmunity in SSc, with strong statistical evidence for IL-6- and NF-κB-associated genes (IL6ST, IRAK1, REL, and TNFAIP3) contributing to cutaneous and vascular pathologies, while variants in interferon and apoptotic regulation genes (SAMD9L, TREX1, and DNASE1L3) may underlie fibrotic and glandular involvement.

The genetic profile of SSc in Kazakh patients mainly aligns with international patterns compared with European, East Asian, and Middle Eastern SSc populations while revealing unique regional variations. Genome-wide association studies from Europe, North America, and East Asia highlight a shared genetic framework concentrated on immune regulation. Of the ~40 known susceptibility loci, 15 show strong cross-population significance, primarily influencing innate and adaptive immunity as well as programmed cell death. Notably, direct genetic links to fibrosis and vascular dysfunction remain limited, mirroring global findings. Variants in immune-regulatory loci that recur in international studies—IRAK1, TNFAIP3, IL6R/IL6ST, and transcriptional regulators such as AFF3 and IKZF3—in our patients are supported by the central role of NF-κB and IL-6 signalling in SSc pathogenesis [37,48]. The proximity of several IRAK1 changes to previously reported functional haplotypes is consistent with associations with diffuse cutaneous disease and ILD reported in European cohorts [2]. Similarly, the TNFAIP3 variant (chr6:138199775 T/TC) supports prior findings in East Asian and Middle Eastern populations implicating A20 haploinsufficiency in heightened NF-κB signalling and chronic inflammation.

Conversely, the enrichment of variants in SAMD9L and ABCC2, and a higher observed frequency of a LY96 (MD-2) variant, appear less well documented in published SSc series and may reflect regional- or admixture-related signals in the Central Asian population (Appendix A). Overall, the Kazakh profile aligns with a trans-ethnic “core” of immune pathway susceptibility (IL-6, NF-κB, and B/T-cell regulators) overlaid by locally enriched variants that could modulate clinical heterogeneity. These insights hold promise for precision medicine, enabling the development of risk models tailored to Kazakh patients.

### Methodological Considerations and Future Directions

The relatively small number of participants should be acknowledged as a limitation of this study. The restricted cohort inevitably constrains statistical confidence, especially in evaluating low-frequency variants or subtle genotype–phenotype links. At the same time, these findings represent an important first step toward understanding the immunogenetic background of SSc in Kazakhstan. The representativeness of this cohort should, however, be interpreted with caution, since it may not fully capture the genetic and clinical diversity of the broader Central Asian population. Expanding this work through national and regional collaborations would provide stronger evidence and help confirm whether the observed trends reflect population-specific features or universal mechanisms of the disease.

Although multiple variants were classified as likely pathogenic according to ACMG criteria, their functional consequences have not been validated in vitro or in vivo. In addition, the cross-sectional design precludes the assessment of whether identified variants predict disease progression or treatment response. Finally, the targeted panel included only coding regions of selected genes and did not capture non-coding or epigenetic mechanisms that may contribute to SSc pathogenesis.

Future studies should therefore expand patient cohorts through national and international collaborations, integrate omics approaches, and apply functional assays to confirm the role of candidate genes such as SAMD9L, IRAK1, TNFAIP3, IL6R, and others mentioned. Establishing a national SSc registry compatible with global databases will be crucial to overcome current sample size limitations. Despite these caveats, our work provides the first genetic snapshot of SSc in the Kazakh population, offering a foundation for precision medicine approaches and future biomarker discovery.

## 5. Conclusions

This study represents the first immunogenetic analysis of SSc in a Kazakh cohort of patients. By integrating antibody profiling and the targeted sequencing of autoimmune-related genes, we identified a set of candidate variants—particularly in SAMD9L, IRAK1, REL, IL6ST, TNFAIP3, ITGA2, ABCC2, AIRE, IL6R/ST, LY96, AFF3, and TREX1—that may contribute to SSc susceptibility and clinical heterogeneity. While the findings are consistent with previously reported pathways of immune dysregulation, they also reveal potential regional specificity in genetic backgrounds.

## Figures and Tables

**Figure 1 pathophysiology-32-00057-f001:**
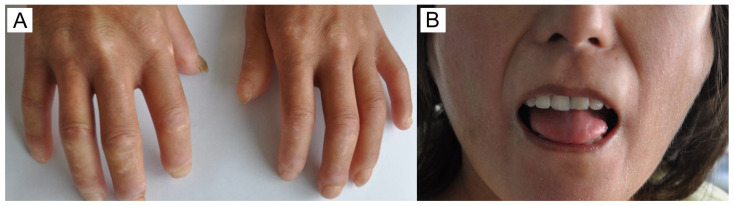
(**A**): Skin impairment in SSc patient: dense edema, digital ulcers, and puffy fingers. (**B**): Perioral soft tissue fibrosis and skin thickness resulted in microstomia.

**Figure 2 pathophysiology-32-00057-f002:**
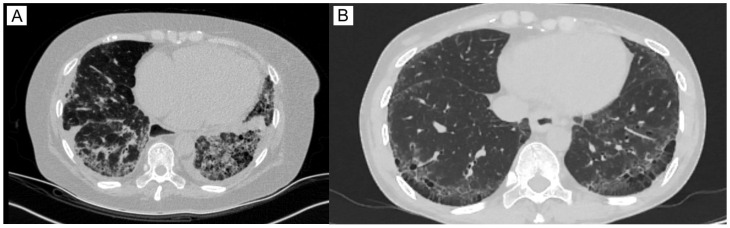
Characteristic features of SSc-ILD: the high-resolution computed tomography axial projection of the chest demonstrates (**A**) signs of non-specific interstitial pneumonia (NSIP), a pronounced reticular pattern with forming honeycombing, and a dilated esophagus lumen; (**B**) signs of usual interstitial pneumonia (UIP), reticular abnormality with traction bronchiectasis, with predominance in the subpleural lower lungs; oesophageal lumen is dilated.

**Figure 3 pathophysiology-32-00057-f003:**
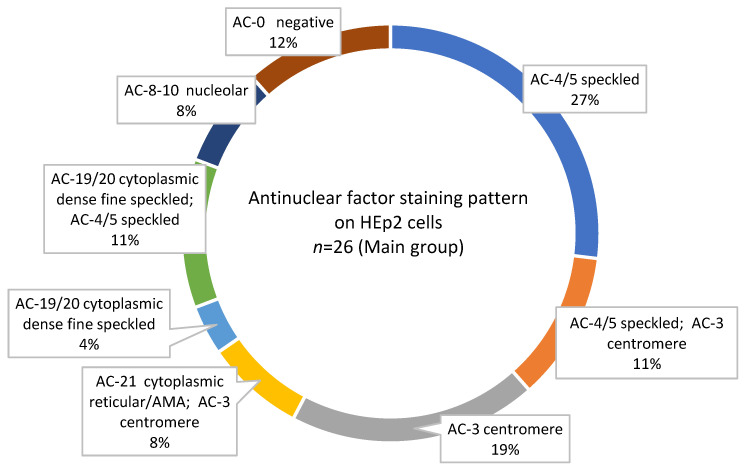
Staining pattern of antinuclear factor analyzed on HEp-2 cells.

**Figure 4 pathophysiology-32-00057-f004:**
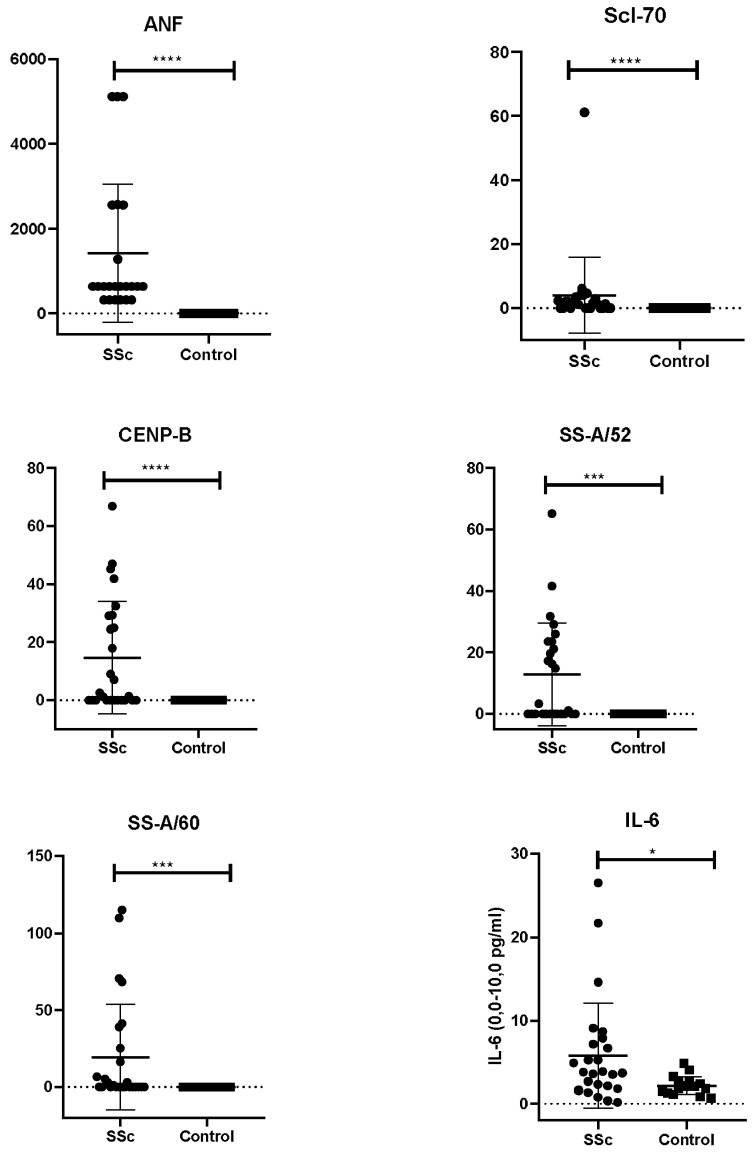
Serum autoantibody and cytokine profiles in Kazakh patients. Scatter plots show serum levels of antinuclear factor (ANF), anti-Scl-70, anti-CENP-B, anti-SS-A/Ro52, anti-SS-A/Ro60 antibodies, and interleukin-6 in SSc patients (*n* = 26) and healthy controls (*n* = 18). Bars represent the mean ± SD. Statistical significance was determined using the Mann–Whitney U test with **** for *p* < 0.0001, *** for *p* < 0.001 and * for *p* < 0.05. Elevated levels of ANF, anti-Scl-70, anti-CENP-B, and SS-A antibodies, as well as IL-6, were observed in the SSc group compared with controls.

**Figure 5 pathophysiology-32-00057-f005:**
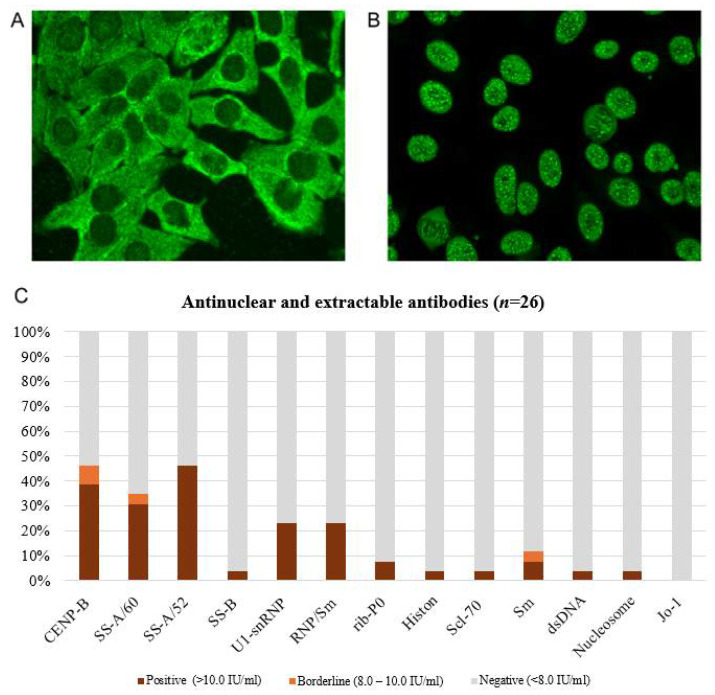
Antinuclear and extractable nuclear antibodies: (**A**) Patient S13, antinuclear factor on HEp-2 cells estimated by indirect immunofluorescence, titre 1:320, cytoplasmic dense fine speckled type (AC-19/20) of glow; positive antibodies to dsDNA, nucleosome, histone, U1-snRNP, SS-A/60, SS-A/52, and ribP0. (**B**) Patient S23, antinuclear factor, titre 1:5120, centromeric type (AC-3) and nuclear speckled type (AC-4/5) of luminescence; positive antibodies to SS-A/60, SS-A/52, and CENP-B. (**C**) Spectrum of antibodies (percentage of ANA/ENA) revealed in SSc patients (reference interval: <8.0 IU/mL is negative, 8.0–10.0 IU/mL is borderline, and >10.0 IU/mL is positive).

**Figure 6 pathophysiology-32-00057-f006:**
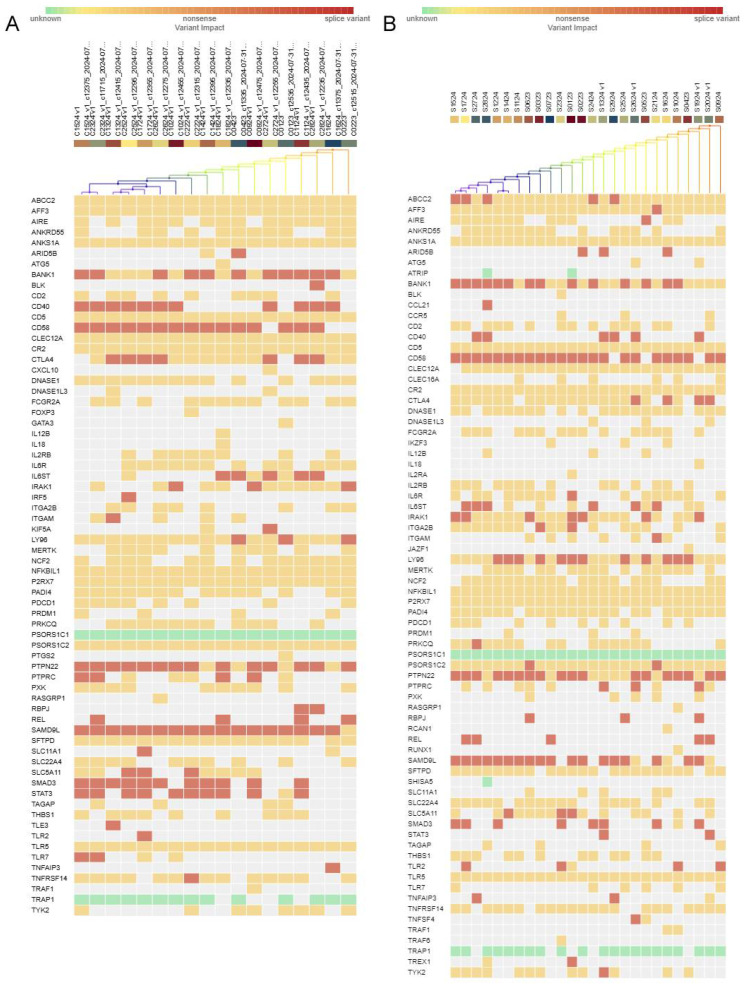
Comparative heatmaps of germline mutations in 120 target genes: (**A**) heatmap for 18 control samples; (**B**) heatmap for 26 systemic sclerosis samples. Rows represent targeted genes, and columns represent individual samples. Samples are hierarchically clustered based on mutation presence and predicted impact, as determined by IonReporter Software (Thermo Fisher Scientific, version 5.20). The colour scale indicates variant classification: green = variant of unknown impact, orange = missense variant, red = nonsense variant, maroon = splice variant, and white = no detected variant (at or above the applied threshold). Hierarchical clustering dendrograms are shown at the top of each heatmap, highlighting groups of samples with similar genetic alteration patterns. A minimum coverage of 30 reads was used as inclusion criteria. Genes on the y-axis are organized according to alphabetical order.

**Table 1 pathophysiology-32-00057-t001:** Antinuclear factor of the main and control groups of patients. Reference interval < 1:80 negative, ≥1:80 positive. Antinuclear factor titre criteria: ≥1:80 (1:80–1:160)—diagnostic (cut-off point); ≥1:160—clinically significant titre.

Antinuclear Factor on HEp-2 Cells, Titer	<1:80 (Negative)	1:320	1:640	1:1280	1:2560	1:5120
Main group (*n* = 26)	3	6	10	1	3	3
Control group (*n* = 18)	18	0	0	0	0	0

**Table 2 pathophysiology-32-00057-t002:** Staining pattern of antinuclear factor analyzed on HEp-2 cells.

Antinuclear Factor Staining Pattern on HEp-2 Cells	Main Group (*n* = 26)	Control Group (*n* = 18)
AC-4/5 speckled	7	0
AC-4/5 speckled; AC-3 centromere *	3	0
AC-3 centromere	5	0
AC-21 cytoplasmic reticular/AMA; AC-3 centromere *	2	0
AC-19/20 cytoplasmic dense fine speckled	1	0
AC-19/20 cytoplasmic dense fine speckled; AC-4/5 speckled *	3	0
AC-8-10 nucleolar	2	0
AC-0 negative	3	18

* Patients in the main group demonstrated combinations of several types of glows.

**Table 3 pathophysiology-32-00057-t003:** Spectrum of detected autoantibodies in the main group (systemic sclerosis patients), including the quantity of patients that demonstrated positive results for antinuclear and extractable antibodies.

SSc, Antinuclear and Extractable Antibodies (ANA/ENA), *n* = 26	Positive (>10.0 IU/mL)	Borderline (8.0–10.0 IU/mL)	Negative (<8.0 IU/mL)
Scl-70	1	0	25
SS-A/60	8	1	17
SS-A/52	12	0	14
CENP-B	10	2	14
U1-snRNP	6	0	20
SS-B	1	0	25
dsDNA	1	0	25
Nucleosome	1	0	25
Sm	2	1	23
RNP/Sm	6	0	20
rib-P0	2	0	24
Histone	1	0	25
Jo-1	0	0	26

**Table 4 pathophysiology-32-00057-t004:** Genetic variants detected in main and control groups.

N	Gene	Locus	Genotype	Occurrence of Variant		Fisher’s Exact Test
				Control Group	Main Group	
1	SAMD9L	chr7:92764981	T/TT	1	3	0.633
		chr7:92761606	GT/G	1	7	0.114
2	LY96	chr8:74922341	CT/C	3	12	0.056
3	REL	chr2:61149099	GT/G	2	5	0.681
4	IRAK1	chrX:153278833	GCC/GCCG	0	3	0.257
		chrX:153278833	GCCCG/GCC	1	3	0.633
5	RBPJ	chr4:26417097	GTTTTTTGC/GTTTTTTTG ref GTTTTTTTGC	0	1	1.0
		chr4:26417097	GT/G	0	1	1.0
		chr4:26426085	C/CT	0	1	1.0
6	IL6ST	chr5:55265588	AT/A	1	2	1,0
		chr5:55265655	G/C	1	2	1.0
7	TNFAIP3	chr6:138199775	T/TC	1	2	1.0
8	ITGA2	chr17:42453072	G/GC (ref GCC)	0	1	1.0
		chr17:42455791	G/A	0	1	1.0
9	ABCC2	chr10:101578956	CA/C	0	2	0.50
		chr10:101603641	CA/C	0	1	1.0
		chr10:101559041	CA/C	0	1	1.0
10	AIRE	chr21:45708278	G/C	0	2	0.50
		chr21:45711068	TC/T	0	1	1.0
		chr21:45713024	A/G	0	1	1.0
		chr21:45711025	C/G	0	1	1.0
11	IL6R	chr1:154378136	GC/G	0	1	1.0
		chr1:154401686	G/A	0	1	1.0
12	JAZF1	chr7:28220153	C/T	0	2	0.50
13	IKZF3	chr17:37922552	T/C	0	2	0.50
14	AFF3	chr2:100623846	GT/G	0	1	1.0
15	TREX1	chr3:48508185	T/TC	0	1	1.0
16	IL18	chr11:112014401	C/T	0	1	1.0
17	IL12B	chr5:158749513	T/C	0	1	1.0
18	PRKCQ	chr10:6527154	AT/A	0	1	1.0
19	PXK	chr3:58385095	T/C	0	1	1.0
20	DNASE1L3	chr3:58183626	G/T	0	1	1.0

## Data Availability

The original contributions presented in this study are included in the article/Appendix A. Further inquiries can be directed to the corresponding author.

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
