# Peer review of "Systemic Sclerosis in Kazakh Patients: A Preliminary Case–Control Immunogenetic Profiling Study"

_pathophysiology, 2025, doi:10.3390/pathophysiology32040057_

Round 1
Reviewer 1 Report
Comments and Suggestions for Authors
In this research article, Zaripova and colleagues performed an immunogenetic analysis of systemic sclerosis in the Kazakh cohort of patients. Alongside identification of specific autoantibodies, their genetic analysis uncovered multiple variants across immune regulatory genes related to systemic sclerosis. This represents the first immunogenetic study conducted on a Kazakhstan Kazakh cohort by providing original insights into the disease genetic and immunological landscape in this population. This study findings could significantly contribute to subsequent follow-up and larger-scale research.
Minor revision suggestions:
The paragraph beginning at line 40 lacks clarity regarding whether the disease incidence data presented pertains specifically to Kazakhstan. Clarification is needed for reader understanding.
In Section 2.2.1 (line 152), please indicate if any major antinuclear factors or specific autoantibodies were not included in this study, and provide justification for their exclusion.
Author Response
We sincerely thank the reviewer for their thoughtful assessment and constructive feedback.
Reviewer’s comment 1: The paragraph beginning at line 40 lacks clarity regarding whether the disease incidence data presented pertains specifically to Kazakhstan. Clarification is needed for reader understanding.
Response: We thank the reviewer for this important observation. For Kazakhstan, specific epidemiological data are limited. Particularly, centralized, comprehensive registry data on the incidence of SSс are still scarce. The incidence and prevalence data cited in the paragraph refer to global and regional estimates reported in the international literature, as currently there are no national registries or epidemiological databases available in Kazakhstan for systemic sclerosis or related autoimmune diseases. We have revised the text to explicitly state that these figures do not represent local data but rather provide context for the global epidemiological variability of systemic sclerosis.
Reviewer’s comment 2: In Section 2.2.1 (line 152), please indicate if any major antinuclear factors or specific autoantibodies were not included in this study, and provide justification for their exclusion.
Response: We appreciate this valuable suggestion. The most common autoantibodies anti-Scl-70, anti-Smith, anti-U1 RNP, anti-SSA/Ro, anti-SSB/La, anticentromere antibodies) were determined in accordance with national clinical protocol and international guidelines. Antibodies to histones, anti-dsDNA and Jo-1 specific for SLE and polymyositis rather than SSc were included because of the same ANA 12 line. Autoantibodies such as anti-Ku, anti-U3 RNP, anti -Th/To, anti-PM/Scl, anti-U11/U12 RNP, anti-PM/Scl and others were not included due to their very low prevalence in systemic sclerosis and the limited availability of standardized assays in our setting. This rationale has been added to the Methods to ensure transparency and reproducibility.
We also improved Methods description, as well as figures and tables as you recommended.
Reviewer 2 Report
Comments and Suggestions for Authors
The study provides important new information from Kazakhstan, but this novelty should be further highlighted. For example, how does this genetic profile compare with findings in cohorts from Europe, East Asia, and the Middle East?
The manuscript should confirm whether the identified variants of uncertain significance (VUS) in LY96, PTPN22, IRAK1, and SAMD9L are newly reported in SSc or have already been described in previous studies.
The abstract is condensed and should clearly distinguish between the objectives, methods, main results, and conclusions.
The supplementary tables are very detailed; however, it is recommended that some extended gene variant descriptions be moved to a separate appendix to simplify the main supplementary material.
The supplementary tables are detailed and informative. However, the patient clinical descriptions are too long and narrative-like. A more structured format (such as standardized codes for organ involvement) would improve readability and reproducibility.
It would be helpful to include statistical comparisons between patient and control groups (IL-6 levels, antibody prevalence, and genetic variants). Currently, supplementary data are descriptive only. 3. Genetic Analysis.
Interpretation of genetic variants is limited. Authors should clarify criteria for classifying variants as "likely pathogenic," "pathogenic," or "nonpathogenic." Reference to ACMG guidelines would enhance the analysis.
The functional implications of key variants (e.g., PTPN22, CTLA4, and REL) should be discussed in greater depth, particularly with respect to known autoimmune pathways.
The manuscript would benefit from statistical evaluation of genotype-phenotype associations (e.g., are certain variants associated with elevated Rodnan skin score or ILD?).
The relatively small sample size should be considered a limitation, and the representativeness of this population should be discussed.
The manuscript is generally understandable, but it contains long and complex sentences that reduce clarity. Editing the language to ensure brevity and fluency is recommended.
Abbreviations (e.g., ILD, PAH, VUS, LP) should be consistently defined when used for the first time in both the main text and the supplementary material.
Ensure that all references cited for specific genes/variants are up to date (2020–2024).
Author Response
We sincerely thank the reviewer for careful evaluation of our manuscript and for the constructive comments that will help us improve the quality and clarity of the paper. Below, we provide point-by-point responses.
Reviewer’s comment 1: The study provides important new information from Kazakhstan, but this novelty should be further highlighted. For example, how does this genetic profile compare with findings in cohorts from Europe, East Asia, and the Middle East?
Response 1: We thank the reviewer for this valuable recommendation. In the revised Discussion, we have expanded the comparison of our cohort’s genetic profile with published data from European, East Asian, and Middle Eastern studies. To estimate the frequency of detected variants in various populations, several international genomic databases were consulted, representing aggregated sequencing data from thousands to hundreds of thousands of individuals of diverse ethnic and geographic origins. gnomAD, dbSNP Database and TOPMed program used to check the frequency of detected variants in various populations (added in text and Supplementary table 3). We emphasize both overlapping risk variants (e.g., PTPN22, CTLA4) and unique patterns, highlighting the novelty of reporting these data from Kazakhstan in LY96, IRAK1 and SAMD9L.
Reviewer’s comment 2: The manuscript should confirm whether the identified variants of uncertain significance (VUS) in LY96, PTPN22, IRAK1, and SAMD9L are newly reported in SSc or have already been described in previous studies.
Response 2: We carefully reviewed the literature PubMed search (cut-off: October 10, 2025) and updated the Results and Discussion. The specific VUS reported in our cohort - LY96 (chr8:74922341 CT/C), PTPN22 (chr1:114381166 CT/C), IRAK1 (indels around chrX:153278833), and SAMD9L (chr7:92761606 GT/G; chr7:92764981 T/TT) - have not been previously reported in SSc cohorts to our knowledge. Prior studies linked SSc risk to other, distinct variants in PTPN22 (rs2476601, R620W) and IRAK1 (rs1059702 and Xq28 haplotype), while LY96/MD-2 involvement has been shown at the level of pathway activation (TLR4-MD2) rather than germline variant association; SAMD9L has no published germline association with SSc. We have clarified this point in the Discussion and Supplementary Tables, including genomic coordinates and references.
Reviewer’s comment 3: The abstract is condensed and should clearly distinguish between the objectives, methods, main results, and conclusions.
Response 3: We appreciate this observation. The abstract has been restructured into four distinct sections (Objectives, Methods, Results, Conclusions) to enhance readability and ensure clarity of the study’s aims and findings.
Reviewer’s comment 4: The supplementary tables are very detailed; however, it is recommended that some extended gene variant descriptions be moved to a separate appendix to simplify the main supplementary material.
Response 4: Thank you for this suggestion. Extended gene variant annotations, type, ACMG classification, as well as population data have been moved into a separate supplementary table 1. The table 4 now focus on core results, improving their usability.
Reviewer’s comment 5: The supplementary tables are detailed and informative. However, the patient clinical descriptions are too long and narrative-like. A more structured format (such as standardized codes for organ involvement) would improve readability and reproducibility.
Response 5: We absolutely agree with this point. Patient clinical data have been reformatted into a structured table using standardized codes for organ involvement (e.g., ILD, PAH, sSS, RP), which make it easier to understand, improving both readability and reproducibility.
Reviewer’s comment 6: It would be helpful to include statistical comparisons between patient and control groups (IL-6 levels, antibody prevalence, and genetic variants). Currently, supplementary data are descriptive only.
Response 6: Thank you, that certainly makes the article more engaging to readers. We have now included statistical comparisons between patient and control groups for IL-6 levels and antibody prevalence. These results are added to the Results section and summarized in Figure 4. According to genetic variants statistical analyses had been done previously, no statistically significant differences were observed between the SSc and control groups. A slight, though not statistically significant, trend was noted for the LY96 genes. This subtle variation may reflect the limited statistical power resulting from the small sample size.
Reviewer’s comment 7: Interpretation of genetic variants is limited. Authors should clarify criteria for classifying variants as "likely pathogenic," "pathogenic," or "nonpathogenic." Reference to ACMG guidelines would enhance the analysis.
Response 7: We thank the reviewer for this suggestion. Variant classification was performed according to ACMG guidelines, and this has been explicitly stated in the Methods.
Reviewer’s comment 8: The functional implications of key variants (e.g., PTPN22, CTLA4, and REL) should be discussed in greater depth, particularly with respect to known autoimmune pathways.
Response 8: We have expanded the Discussion to include a more detailed functional interpretation of these variants, integrating evidence from autoimmune signaling pathways, including T-cell receptor regulation, co-stimulation, and NF-κB activation. Together, PTPN22, CTLA4, and REL represent successive checkpoints along a shared autoimmune pathway: modulation of TCR activation, control of costimulatory feedback, and transcriptional regulation of effector responses. Their combined disturbance sustains lymphocyte activation, weakens tolerance mechanisms, and fosters continuous cross-talk between immune and stromal compartments. This convergence links inherited susceptibility with the clinical manifestations of fibrosis and vascular injury that define systemic sclerosis.
Reviewer’s comment 9: The manuscript would benefit from statistical evaluation of genotype-phenotype associations (e.g., are certain variants associated with elevated Rodnan skin score or ILD?).
Response 9: We conducted exploratory analyses of genotype–phenotype correlations, focusing on different organs and tissues impairment. Although limited by sample size, preliminary associations are now presented in Results and Discussion.
Reviewer’s comment 10: The relatively small sample size should be considered a limitation, and the representativeness of this population should be discussed.
Response 10: We agree and have added a section in the Discussion acknowledging the limitation of sample size and the potential challenges in generalizability. The restricted cohort inevitably constrains statistical confidence, especially in evaluating low-frequency variants. At the same time, we emphasize the value of these first genetic data from the Kazakh population.
Reviewer’s comment 11: The manuscript is generally understandable, but it contains long and complex sentences that reduce clarity. Editing the language to ensure brevity and fluency is recommended.
Response 11: The manuscript has been thoroughly revised and edited. Sentences have been shortened and simplified to improve clarity and fluency.
Reviewer’s comment 12: Abbreviations (e.g., ILD, PAH, VUS, LP) should be consistently defined when used for the first time in both the main text and the supplementary material.
Response 12: We reviewed all abbreviations throughout the text and supplementary material to ensure that each is defined at first mention and used consistently thereafter.
Reviewer’s comment 13: Ensure that all references cited for specific genes/variants are up to date (2020–2024).
Response 13: References have been updated to include the most recent publications from 2020–2024 relevant to the discussed variants and pathways.
We have carefully revised the manuscript to address the points noted as requiring improvement. We have clarified the methodology, the results section with concise descriptions of key findings; the tables have been reformatted to enhance readability. We added Figure 4 with antibodies' statistics between main and control group and supplementary tab.3, with uniform labeling, improved legends, and clearer presentation of statistical significance.
We believe these revisions have significantly strengthened the overall scientific quality, transparency, and readability of the manuscript.
Round 2
Reviewer 2 Report
Comments and Suggestions for Authors
The authors have submitted a revised version of their manuscript, accompanied by a detailed and concise response to the comments of the previous review. They have made a diligent and satisfactory effort to address most of the concerns raised. The manuscript is significantly improved, with enhanced clarity, a more balanced interpretation of the results, and better contextualization within the global literature. The inclusion of population frequency data from international genomic databases and a more careful framing of genetic associations are particularly commendable. This work represents a valuable and novel contribution to the field of SSc genetics in an understudied population.
-The extensive discussion comparing Kazakh genetic profiles with cohorts from Europe, East Asia, and the Middle East, supported by data from gnomAD, dbSNP, and TOPMed, effectively highlights the novelty and regional specificity of the study.
-The authors convincingly demonstrate through a literature review that the identified variants in LY96, PTPN22, IRAK1, and SAMD9L reported here have not previously been associated with amyotrophic scleroderma, further strengthening the impact of these findings.
- The restructuring of the abstract and reformatting of the supplementary tables (especially the clinical data) have significantly enhanced the organization and accessibility of the manuscript.
- The addition of statistical comparisons for IL-6 and autoantibodies, along with the explicit acknowledgement of the limited strength of genetic associations, provides a more transparent and accurate presentation of the results.
-Explicit reference to ACMG/AMP guidelines for variant classification and in-depth discussion of the functional effects of key genes (PTPN22, CTLA4, REL) strengthen the methodological and interpretive framework.
Although the manuscript is much stronger, there are a few minor points that require attention to ensure a revised final version:
1. The authors state that the variants in LY96, PTPN22, IRAK1, and SAMD9L are "newly reported." This is true for specific genomic coordinates (e.g., chr8:74922341). However, to avoid any potential misunderstanding, it would be prudent to explicitly state that these variants are present (and have not been previously reported) in these genes, to distinguish them from known variants associated with SSc in the same genes (e.g., the known PTPN22 rs2476601). In the Abstract and Discussion, they used phrases such as "previously unreported variants in the LY96 genes..." to ensure complete clarity.
2. The authors added an appropriate limitations section. It is essential that the language of the Abstract's conclusion reflects the cautious tone of the main text. Be sure to include a phrase such as "...although the limited sample size and lack of functional validation limit the interpretability of the results..." to set the right expectations from the outset.
3. A thorough final proofread is recommended to detect any minor grammatical or typographical errors that may remain after extensive revisions. Pay close attention to new text and revised sentences. Make careful line-by-line edits before final submission.
The authors were very responsive to the initial review and successfully implemented changes that significantly improved the quality, clarity, and scientific rigor of the manuscript. The study provides important baseline data on SSc in the Kazakh population and identifies interesting candidate genes for future research. The remaining revisions are minor and can be addressed quickly. After incorporating these final points, the manuscript will be ready for publication.
Author Response
Thank you for the careful reading and constructive suggestions. We have addressed each point in the revised manuscript; below is a concise point-by-point response that we will include in the formal rebuttal.
Reviewer comment 1: The authors state that the variants in LY96, PTPN22, IRAK1, and SAMD9L are "newly reported." This is true for specific genomic coordinates (e.g., chr8:74922341). However, to avoid any potential misunderstanding, it would be prudent to explicitly state that these variants are present (and have not been previously reported) in these genes, to distinguish them from known variants associated with SSc in the same genes (e.g., the known PTPN22 rs2476601). In the Abstract and Discussion, they used phrases such as "previously unreported variants in the LY96 genes..." to ensure complete clarity.
Authors’ response:
We agree that precision is essential to avoid misinterpretation. We revised the Abstract, Results, and Discussion to explicitly state that the listed changes are novel at the reported genomic coordinates and that they are distinct from previously described SSc-associated polymorphisms in the same genes. In the Abstract and Discussion we changed the phrase “…previously unreported variants in the LY96 genes…” to:
“Novel variants at the following genomic coordinates were identified and have not been previously reported in association with SSc: LY96 (chr8:74922341 CT/C), PTPN22 (chr1:114381166 CT/C), IRAK1 (indels at chrX:153278833), and SAMD9L (chr7:92761606 GT/G; chr7:92764981 T/TT).”
Reviewer comment 2: The authors added an appropriate limitations section. It is essential that the language of the Abstract's conclusion reflects the cautious tone of the main text. Be sure to include a phrase such as "...although the limited sample size and lack of functional validation limit the interpretability of the results..." to set the right expectations from the outset.
Authors’ response:
We have revised the final sentence of the Abstract to reflect a cautious interpretation:
“Although the limited sample size and lack of functional validation constrain the interpretability of the findings...”
This wording mirrors the limitations described in the Discussion and aligns readers’ expectations from the outset.
Reviewer comment 3: A thorough final proofread is recommended to detect any minor grammatical or typographical errors that may remain after extensive revisions. Pay close attention to new text and revised sentences. Make careful line-by-line edits before final submission.
Authors’ response:
We conducted careful line-by-line proofreading of the entire manuscript, with particular attention to newly added and revised passages. Minor typographical and grammatical errors were corrected.